# A Benzothiazole-Based Fluorescent Probe for Ratiometric Detection of Al^3+^ and Its Application in Water Samples and Cell Imaging

**DOI:** 10.3390/ijms20235993

**Published:** 2019-11-28

**Authors:** Zhen-Nan Tian, Ding-Qi Wu, Xue-Jiao Sun, Ting-Ting Liu, Zhi-Yong Xing

**Affiliations:** Department of Applied Chemistry, College of Arts and Sciences, Northeast Agricultural University, Harbin 150030, China

**Keywords:** benzothiazole, ratiometric, fluorescent probe, Al^3+^

## Abstract

An easily prepared benzothiazole-based probe (BHM) was prepared and characterized by general spectra, including ^1^H NMR, ^13^C NMR, HRMS, and single-crystal X-ray diffraction. Based on the synergistic mechanism of the inhabitation of intramolecular charge transfer (ICT), the BHM displayed high selectivity and sensitivity for Al^3+^ in DMF/H_2_O (*v/v*, 1/1) through an obvious blue-shift in the fluorescent spectrum and significant color change detected by the naked eye, respectively. The binding ratio of BHM with Al^3+^ was 1:1, as determined by the Job plot, and the binding details were investigated using FT-IR, ^1^H NMR titration, and ESI-MS analysis. Furthermore, the BHM was successfully applied in the detection of Al^3+^ in the Songhua River and on a test stripe. Fluorescence imaging experiments confirmed that the BHM could be used to monitor Al^3+^ in human stromal cells (HSC).

## 1. Introduction

Aluminum, as one of the most abundant metals in the earth, is widely used in daily life, including in kitchen utensils, pharmaceutical packaging, and food additives [1,2,3,4]. These activities will inevitably induce the accumulation of Al^3+^, which is the most ionic style of aluminum in existence in environmental water and the human body. However, as a non-necessary element for humans, excess accumulation of Al^3+^ in the human body might be associated with diseases, including Alzheimer’s disease, Parkinson’s disease, amyotrophic sclerosis, and encephalopathy [5,6,7,8,9,10]. According to the regulations of the World Health Organization (WHO), the maximum concentration of Al^3+^ is 7.4 μM in drinking water. Hence, it is necessary to construct a simple and effective method with the function of qualitative and quantitative analysis in Al^3+^ detection.

The development of fluorescent probes for detection and special analysis has attracted many researchers, and numerous excellent probes for Al^3+^ have been reported [11,12,13,14,15,16,17,18,19,20,21,22,23,24,25,26,27]. Among them, the ratiometric probe is one of the most popular types of sensors due to the inherent advantage that it can eliminate the interference of the external environment through self-calibration of its response signals either in emission or absorbance intensities at two different wavelengths. Some ratiometric Al^3+^ probes were developed using different fluorophores, including rhodamine–naphthalene [17], rhodamine-benzothiazole [18], 4′-methyl-3-hydroxyflavone [19], coumarin-quinoline [20], rhodamine-chromone [21], carbazole-rhodamine [22], imidazoquinazoline [23], benzothiazole [24], benzimidazole [25], naphthalimide [26], and quinoline–coumarin [27]. However, only a few of these were benzothiazole-based fluorescent probes for the ratiometric detection of Al^3+^ [24]. Moreover, benzothiazole, an excellent fluorophore due to its distinguished merits, such as excellent photo stability and easy structural modification, had been used in the construction of chemosensors for the detection of various ions [28,29,30,31,32,33,34]. Moreover, this type of structure, in which the hydroxyl group acts as an electron donating unit in the para-position of the benzothiazolyl group used as the electron withdrawing group, was facile for the ICT-based mechanism by increasing the push-pull electron effect [35]. Given the above statements, a novel easily prepared probe BHM (5-(benzo[d]thiazol-2-yl)-2-hydroxy-3-methoxybenzaldehyde) was synthesized, and its stereo structure was confirmed by single crystal X-ray diffraction (Scheme 1). The BHM showed an obvious selectivity and sensitivity to Al^3+^ with a significant blue-shift both in the emission spectrum and absorbance spectrum. Its application in real water samples was investigated. Moreover, the BHM was successful in imaging the human stromal cell line (HSC).

## 2. Results and Discussion

### 2.1. Synthesis of BHM

The BHM was facile synthesized through a Duff reaction using the starting material of 4-(benzo[d]thiazol-2-yl)-2-methoxyphenol (3), which was synthesized via a condensation reaction between 2-aminobenzenethiol and 4-hydroxy-3-methoxybenzaldehyde [36]. The single crystal of BHM was obtained, and its molecular structure is depicted in Figure 1. The selective crystal data are displayed in Appendix A.

### 2.2. Spectra Performance Investigation of the BHM to Metal Ions

The selectivity of BHM (10 μM) to different metal ions (Na^+^, K^+^, Mg^2+^, Ca^2+^, Ag^+^, F e^2+^, Co^2+^, Ba^2+^, Mn^2+^, Cu^2+^, Zn^2+^, Cd^2+^, Hg^2+^, Ni^2+^, Pb^2+^, Cr^3+^, Fe^3+^, and Al^3+^) was firstly determined using UV-Vis spectrum in a solution of DMF:H_2_O (1:1, *v/v*) (Figure 2). The UV-Vis absorbance of BHM showed almost no change after the addition of the tested metal ions, except for Al^3+^ in DMF:H_2_O (1:1, *v/v*) (Figure 2a). As shown in Figure 2a, upon the addition of Al^3+^, an obvious blue-shift from an absorbance peak at 359 to 321 nm was observed compared to that of the BHM itself, which indicated the existence of a strong interaction between the BHM and Al^3+^. Furthermore, the absorbance titration result (Figure 2b) showed the absorbance centered at 359 nm was gradually decreased while the peak at 321 nm was gradually merged with the isosbestic point at 335 nm, indicating the formation of a BHM-Al^3+^ complex. A good relationship was detected between the absorbance ratios (A_307_:A_359_) (Appendix A) and the Al^3+^ concentration (0–12 μM) with a limit of detection (LOD) as low as 99 nM, explaining ratiometric detection by BHM of Al^3+^.

Moreover, the selectivity of BHM (10 μM) to the different metal ions mentioned above was also determined by the fluorescence spectrum in the solution of DMF:H_2_O (1:1, *v/v*) (Figure 3). The result showed that some ions, including Ni^2+^, Pb^2+^, Hg^2+^, Fe^2+^, Fe^3+^, Cr^3+^, and Cu^2+^, decreased the fluorescence intensity of BHM to different extents. However, the addition of Al^3+^ caused a significant blue-shift from 527 to 478 nm, accompanied by a color change from green to deep sky-blue (Figure 3a). This result indicated that the BHM had high selectivity to Al^3+^. Fluorescent titration (Figure 3b) through the gradual addition of Al^3+^ into the BHM solution DMF:H_2_O (1:1, *v/v*) showed that a significant blue-shift of 47 nm was observed, which might be attributed to the depression of the ICT process after coordination with the Al^3+^ [37]. Moreover, the good relationship between the fluorescent intensity (E_m_ = 478 nm) of BHM (10 μM) and the Al^3+^ concentration (12–28 μM) (Appendix A) achieved a LOD (limit of detection was calculated using 3σ/k, where σ is the standard deviation of the blank measurements, and k is the slope of the intensity ratio versus the sample concentration plot) as low as 4.39 μM calculated according to previous methods [35,38].

A competition experiment was conducted to verify the ability of BHM under conditions of disturbance coming from other co-existing metal ions. As illustrated in Figure 4, only metal ions (including Cr^3+^ and Mn^2+^) disturbed the detection of Al^3+^.

### 2.3. Sensing Mechanism of the BHM to Al^3+^

A Job plot was firstly carried out to determine the binding ratio between BHM and Al^3+^ (Figure 5). The result showed that the fluorescence intensity (recorded at 478 nm) of BHM reached a maximum when the proportion of BHM in the total concentration of BHM and Al^3+^ was 0.5, indicating that the binding ratio was 1:1 between BHM and Al^3+^. This result was further supported by the HRMS analysis displayed in Figure 6. Peaks at *m/z* 446.0606 and m/z 464.0116 were attributed to [BHM- H^+^ + Al^3+^ + NO_3_^−^ + DMF]^+^ (Calcd: *m/z* 446.0603) and [BHM- H^+^ + Al^3+^ + NO_3_^−^ + DMF + H_2_O]^+^ (Calcd: *m/z* 464.0708), respectively. Moreover, the binding constants of BHM and Al^3+^ were 1.88 × 10^4^ M^−1^ (Appendix A) and 7.49 × 10^3^ M^−1^ (Appendix A) based on the Benesi–Hilderbrand plot [39,40] recorded by the fluorescence and absorbance signals, respectively.

For explicit binding sites of BHM with Al^3+^, we measured the FT-IR spectrum of free BHM and complex BHM–Al^3+^ (Figure 7), respectively. The typical stretching vibration peaks of the BHM itself located at 3303 cm^−1^ and 1655 cm^−1^ were designated to the hydroxyl group (-OH) and carbonyl group (-C=O), respectively. For the FT-IR spectrum of the BHM–Al^3+^, the stretching bands of -OH turned into a broad peak, and the stretching band of C=O decreased significantly and shifted from 1655 cm^−1^ to 1624 cm^−1^. These results indicated that the coordination of Al^3+^ with oxygen atoms came from the hydroxyl group and carbonyl group, respectively.

^1^HNMR titration experiments were measured in DMSO-*d_6_* to determine the binding sites of BHM with Al3+ further. The proton signals of the BHM itself were analyzed and are shown in Figure 8. After the addition of Al^3+^, the proton signal of hydroxyl (H_a_) located at 10.96 ppm gradually disappeared, suggesting the deprotonation on hydroxyl during the combination of BHM with Al^3+^, in accordance with the result obtained by the FT-IR analysis.

Hence, according to the experiment results (including the Job plot, ^1^HNMR titration, and HRMS) mentioned above, the probable sensing mechanism is illustrated in Scheme 2.

## 3. Materials and Methods

### 3.1. Materials and Apparatus

All analytical reagent grade chemicals and solvents employed for the synthesis and characterization were procured from commercial sources and used as received without any treatment. Nuclear magnetic resonance (NMR) spectra were recorded on a Bruker 600 MHz (Beijing, China) system, and the chemical shifts were reported in ppm with Me_4_Si as the internal standard. High-resolution mass spectroscopy (HRMS) was carried out with negative ion modes on a Waters Xevo UPLC/G2-SQ Tof MS spectrometer (Shanghai, China). The FT-IR spectra were recorded on a Bruker ALPHA-T (Beijing, China) by dispersing samples in KBr disks, in the range of 4000–400 cm^−1^. The UV–Vis absorption and fluorescence spectra of the samples were measured on a Pgeneral TU-2550 UV-Vis Spectrophotometer (Suzhou, Jiangsu, China) and Perkin Elmer LS55 fluorescence spectrometer (Shanghai, China), respectively.

### 3.2. General Information

Stock solutions of the metal ions (Al^3+^, Fe^3+^, Cr^3+^, Ca^2+^, Pb^2+^, Cd^2+^, Cu^2+^, Co^2+^, Zn^2+^, Fe^2+^, Mn^2+^, Mg^2+^, Ni^2+^, Hg^2+^, Na^+^, K^+^, Ag^+^) from the nitrate, chloride, or perchlorate salts were prepared with ultrapure water. BHM (0.1 mM) was dissolved in DMF, which was then diluted by adding ultrapure water to 10 μM. The diluted solution DMF:H_2_O (1:1, *v/v*) was used for the spectra measurement in UV-Vis and fluorescence. The excitation wavelength (E_x_ = 380 nm) was used for the fluorescence experiments, and the slits, including excitation and the emission, were all set to 10 nm.

### 3.3. Synthesis

Compounds 3 and BHM were synthesized according to the methods described in References [34,36].

#### 3.3.1. Synthesis of Probe BHM (5-(benzo[d]thiazol-2-yl)-2-hydroxy-3-methoxybenzaldehyde)

The mixture of compound 3 (500 mg, 1.94 mmol) and hexamethylenetetramine (1.37 g, 9.75 mmol) in trifluoroacetic acid (15 mL) was refluxed for 8 h. After the completion of the reaction, distilled water (30 mL) was added and then stirred for 1 h. The precipitate was filtered off and further purified by silica gel column chromatography using CH_2_Cl_2_:CH_3_OH (20:1, *v/v*) as eluent to obtain the compound BHM (359 mg, 1.26 mmol). Yield: 65%. m.p.: 219.8–220.5 °C.

^1^H NMR (600 MHz, DMSO-d_6_) (Appendix A) δ (ppm) 10.96 (s, 1H), 10.37 (s, 1H), 8.13 (d, J = 7.8 Hz, 1H), 8.04 (d, J = 8.4 Hz, 1H), 7.88 (s, 1H), 7.86 (s, 1H), 7.54 (t, J = 7.2 Hz, 1H), 7.52 (t, J = 7.8 Hz, 1H), 4.02 (s, 3H). ^13^C NMR (151 MHz, DMSO-d_6_) (Appendix A) δ (ppm) 191.08, 166.95, 153.96, 153.90, 149.69, 134.79, 127.12, 125.83, 124.67, 123.11, 123.08, 122.75, 119.56, 114.47, 56.86. HRMS (m/z) (TOF MS ES^-^, Negative Ion Modes) (Appendix A): calcd for C_15_H_10_NO_3_S^-^: 284.0381 [M-H^+^]^-^, found: 284.0385.

#### 3.3.2. Preparation of the Crystal BHM

BHM (285 mg, 0.1 mmoL) was dissolved in ethanol (10 mL), and the mixture was refluxed for 1 h. After cooling to room temperature, the solution was filtered, and the filtrate was kept at room temperature for 3 days to obtain single crystals of BHM that were suitable for X-ray analysis.

### 3.4. Cell Culture and Staining

The fibroblast cell line, human stromal cell line (HSC), was purchased from ATCC (CRL-4003). For maintained HSC, the cells were routinely cultured in mixture medium (DMEM: F-12 = 1:1) that was supplemented with 10% heat-inactivated FBS, 100 U/mL penicillin, 100 μg/mL streptomycin, and 1 mM sodiumpyruvate at 37 °C, 5% CO_2_. The HSC was placed in 6-well plates with sterile cover glass at a concentration of 105 cells/well; and after 48 h, the media was changed to being without FBS or antibiotics for chemical treatment. These cells were incubated for 2 h using different amounts of Al^3+^ (10 and 100 μM). Then, fibroblast cells were washed with D-Hank’s salt solution 3 times. Before staining with BHM, the cells were fixed using a standard paraformaldehyde fixation protocol and then rinsed with 5:5 mixture solutions of DMF and water. Then, the cells were stained by incubating for 2 h with BHM (1 × 10^−5^ M). Lastly, the cover glass was mounted over slide glass with an anti-fluorescence quenching agent and imaged using fluorescence microscopy.

### 3.5. X-Ray Diffraction Studies

The data collections of the single crystal X-ray diffraction of BHM (0.24 × 0.23 × 0.2 mm) were carried out on a Rigaku AFC-10/Saturn 724 + CCD diffractometer with graphite-monochromated Mo Kα radiation (λ = 0.71073 Å), using the multi-scan technique. The structures were determined by direct methods using SHELXS-2014 and refined by full-matrix least-squares procedures on F^2^ with SHELXL-2014. A total of 9673 integrated reflections were collected, and 2896 were unique in the range with R_int_ = 0.0434 and R_sigma_ = 0.0409. The maximum/minimum residual electron density = 0.396/−0.428 eÅ^−3^. Structural information was deposited with the Cambridge Crystallographic Data Center (CCDC 1902058).

## 4. Applications

### 4.1. Application in Water Samples

BHM for the determination of Al^3+^ was carried out in a real water sample to verify its practical application. The water samples were collected from the campus of our university and the Songhua River in Heilongjiang Province. Then, Al^3+^ at different levels (15–20 μM) was mixed with the water samples. The fluorescence responses of the BHM were recorded for sensing Al^3+^ at 478 nm (Figure 8). The results showed that with the increase in the concentration of Al^3+^ in the tested samples (including ultrapure water, tap water, and Songhua water), the fluorescence intensity increased gradually, and good linearity was found between the fluorescence intensity and Al^3+^ over the concentration range of 15–20 μM (Appendix A). The desirable recovery and relative standard deviation (RSD) values (Table 1) indicated that BHM could be applied in the quantitative analysis of real water samples for the detection of Al^3+^.

### 4.2. Application in Cell Imaging

Monitoring metal ions in biological systems is an important aspect in the evaluation of a probe’s application ability. Therefore, human stromal cells (HSC) were employed for incubation using different amounts of Al^3+^ and BHM (10 μM) for a certain time and then imaged using fluorescence microscopy. As shown in Figure 9A, the cells displayed green fluorescence after incubation with BHM (10 μM) for 2 h. However, after incubation with Al^3+^ (50 μM) and BHM (10 μM), weak blue fluorescence was detected (Figure 9B). The blue fluorescence intensity increased when the cells were treated with a higher concentration of Al^3+^ (100 μM) in the presence of the same concentration of BHM (10 μM) (Figure 9C). These results indicated that BHM could be used as an Al^3+^ monitor in biological systems.

## 5. Conclusions

In summary, an easily prepared benzothiazole-based probe BHM was prepared and characterized. The BHM displayed high selectivity and sensitivity for Al^3+^ in DMF:H_2_O (1:1, *v/v*) through an obvious blue-shift in the fluorescent spectrum and significant color change detected by the naked eye, respectively. The binding ratio of BHM with Al^3+^ was 1:1, as determined by the Job plot, where the binding details were investigated by FT-IR, ^1^H NMR titration, and ESI-MS analysis. Furthermore, BHM was successfully applied in the detection of Al^3+^ in the Songhua River, on a test stripe, and in human stromal cells (HSC).

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
