# Peer review of "A Benzothiazole-Based Fluorescent Probe for Ratiometric Detection of Al3+ and Its Application in Water Samples and Cell Imaging"

_ijms, 2019, doi:10.3390/ijms20235993_

Round 1

Reviewer 1 Report

The authors synthesized a novel benzothiazole derivative (BHM) which functions as a ratiometric colorimetric and fluorometric sensor for Al(3+) ions. The applicability of BHM in water analysis and in fluorescence imaging are demonstrated. These results may be worth for a publication in IJMS, the manuscript, however requires revision at a number of points.

Most of Al(3+) salts are relatively harmless, the sentence in Introduction on the diseases inducible by the excess of Al ions is misleading - it may be deleted If the authors found papers on fluorescent Al(3+) sensors with the same coordination sphere as in BHM, these studies should be cited. The experimental details of the X-ray diffraction studies in Section 3. 1 would fit rather to section ‘2. 1 Materials and apparatus’ I suggest to reverse the order of the discussions of the absorption and fluorescence spectra in Section 3.2. Presumably, the absorption maximum of the Al-BHM complex was selected as the excitation wavelength in the fluorescence measurements. The formula used for the calculation for the determination of the LOD values should be presented. I wonder what LOD means for a sigmoid shaped titration curve shown in the inset of Fig. 2b. The spectra obtained at 0 and the maximal Al concentrations in Fig. 3b look identical as the two spectra in Fig 3a. Fig 3a may be redundant. ‘Ex’ and ‘Em’ appear in the text as well as in the figures repeatedly, instead of lambdaEx and lambdaEm.

Author Response

Thanks the Reviewer #1 for the suggestions about our manuscript. All revised parts were labeled in red color in revised manuscript.

1. According to many reported manuscript illustrated as followed, excess accumulation in human body might induced some disease. So, the expression in introduction was revised as “Al3+ as for the non-necessary element for mankind, the excess accumulation in human body might associate with some diseases including Alzheimer’s disease and Parkinson’s disease, amyotrophic sclerosis and encephalopathy”.

Ref: (1) A novel ratiometric fluorescent probe for selective detection of Hg2+, Cr3+ and Al3+ and its bioimaging application in living cells, Sensors and Actuators B 253 (2017) 1055–1062.

(2) Substituent effect: A new strategy to construct a ratiometricfluorescent probe for detection of Al3+ and imaging in vivo, Sensors and Actuators B 264 (2018) 304–311.

(3) A novel FRET ‘off–on’ fluorescent probe for the selective detection of Fe3+, Al3+ and Cr3+ ions: Its ultrafast energy transfer kinetics and application in live cell imaging, Biosensors and Bioelectronics 68 (2015) 749–756.

2. To our best knowledge, fluorescent Al3+ sensor with the same coordination sphere as in BHM was scarcely reported. So, there was no reference could be cited.

3. The experimental details of the X-ray diffraction studies in Section 3. 1 was added to section ‘2. 1 Materials and apparatus’. Please see it on page 3 in revised version.

4. The order of the discussions of the absorption and fluorescence spectra in Section 3.2. were reversed. So, the selectivity of BHM to different metal ions through the absorption spectra was added on page 3 in revised version.

5. Yes, the absorption maximum of the Al-BHM complex was selected as the excitation wavelength in the fluorescence measurements.

6. The detection limits (calculated using 3σ/k, where σ is the standard deviation of the blank measurements, and k is the slope of the intensity ratio versus sample concentration plot). This part was added on page 3 in revised version.

7. Fig 3a was deleted. Please see it in revised version.

8. ‘λex’ and ‘λem’ were all replaced by ‘Ex’ and ‘Em’ throughout the revised manuscript.

Reviewer 2 Report

Tian Z-N. et al. report a novel small molecule-based on benzimidazole for the detection Al (III). The small molecule has been thoroughly characterized such as 1H NMR, x-ray crystallography etc. Variety of experiments have been carried out that unequivocally prove the probe’s selectivity toward Al3+ ions.

The following points need to be considered prior to the publication.

BHM should show have 11 proton signals only but it was written 12 protons (line 78 to 80). Is the excitation wavelength (Ex = 420 nm) or 380 nm? (line 68) Figure 4: Experiments need to be triplicated. The Bar graph should include the error bars (S.D. of triplicate samples (at least)) Inset Figure located in Figure 2B is small and unclear, making it difficult to understand. Enlarge Figure-inset into the same size as Figure 2B or 2A. ESI-MS: About Figure 6, Is negative ionization mode used? If so, please mention it. Experimental part for the cell imaging is missing. What is “Z” in Figure legend 9? Scale bar of Fluorescence micrographs (Figure 9) should be written in the Figure legend. As some cell lines are sensitive to DMSO concentration, mentioning its final concentration is important. I suspect a high concentration of DMSO might be detrimental to human stromal cells (HSC) cells. Line 147 and 148: It was mentioned “1.88×104 147 M-1 (Figure S6) and 7.49×103 M-1 (Figure S7)”. Authors should explain as to how they extracted these values from Figure S6 and S7. It appears that slope values were not used. The rationale for the design and use of benzothiazole should be justified explicitly. Typographical: cellsincubation (line 199) etc.

Author Response

Thanks the Reviewer #2 for the suggestions on improving our manuscript. All revised parts were labeled in red color in revised manuscript.

The analysis of 1H NMR of BHM was revised. Please see it on page 7 in revised version. The excitation wavelength was 380 nm, the typing error was revised on page 7 in revised version. The error bars was added in the Figure 4 on page 4 in revised version. Inset Figure located in Figure 2B was enlarged enough which could be seen clearly and shown as Figure 3 because the sequence of Figure 2 and 3 was reversed. The negative ionization mode was used which was added on page 7 in revised version. Experimental part for the cell imaging was added on page 7 in revised version. It is typing error. “Z” was replaced by BHM on page 9 in revised version. Scale bar of Fluorescence micrographs (Figure 9) was added in the Figure legend on page 9 in revised version. As for cell imaging experiments, the solvent we used is the mixture solutions of DMF and water (1:1, v/v). Please see it on page 7 in revised version. The binding constant of BHM and Al3+ were 1.88×104 M-1 (Figure S6) and 7.49×103 M-1, which were calculated by the equation:. Where, A is the Intercept of the plot; B is the Slope of the plot. The rationale for the design and use of benzothiazole was added on page 2 in revised version. Typographical error was revised on page 9 in revised version.

Round 2

Reviewer 2 Report

Authors have provided a substantially improved version of the manuscript that addresses all the concerns raised by reviewers. The manuscript is recommended for publication in IJMS.

Author Response

Dear reviewer,

The text has been checked for correct use of grammar and common technical terms by English language editing of MDPI. Please see the attachment for the certificate file. Moreover, all revised parts were labeled in red color in revised manuscript. Thanks a lot for the suggestions on improving our manuscript.